# Efficacy of a Robot-Assisted Intervention in Improving Learning Performance of Elementary School Children with Specific Learning Disorders

**DOI:** 10.3390/children9081155

**Published:** 2022-07-31

**Authors:** Maria T. Papadopoulou, Elpida Karageorgiou, Petros Kechayas, Nikoleta Geronikola, Chris Lytridis, Christos Bazinas, Efi Kourampa, Eleftheria Avramidou, Vassilis G. Kaburlasos, Athanasios E. Evangeliou

**Affiliations:** 14th Department of Pediatrics, Papageorgiou General Hospital, Aristotle University of Thessaloniki, 56403 Thessaloniki, Greece; aeevange@auth.gr; 2Child & Parent Center S.A., Center for Special Education, 54351 Thessaloniki, Greece; elpis_kar@hotmail.com (E.K.); kourampa@hotmail.com (E.K.); 31st Psychiatric Clinic, Papageorgiou General Hospital, Aristotle University of Thessaloniki, 56403 Thessaloniki, Greece; pekechag@gmail.com (P.K.); avramidouria@yahoo.gr (E.A.); 4Euroaction S.A., 54655 Thessaloniki, Greece; nikoletta.geronikola@gmail.com; 5HUMAIN-Lab, Department of Computer Science, International Hellenic University, 65404 Kavala, Greece; lytridic@cs.ihu.gr (C.L.); chrbazi@cs.ihu.gr (C.B.); vgkabs@teiemt.gr (V.G.K.)

**Keywords:** special education, Social Robots (SRs), humanoid robot NAO, Specific Learning Disorders (SpLDs), reading comprehension, phonological awareness

## Abstract

(1) Background: There has been significant recent interest in the potential role of social robots (SRs) in special education. Specific Learning Disorders (SpLDs) have a high prevalence in the student population, and early intervention with personalized special educational programs is crucial for optimal academic achievement. (2) Methods: We designed an intense special education intervention for children in the third and fourth years of elementary school with a diagnosis of a SpLD. Following confirmation of eligibility and informed consent, the participants were prospectively and randomly allocated to two groups: (a) the SR group, for which the intervention was delivered by the humanoid robot NAO with the assistance of a special education teacher and (b) the control group, for which the intervention was delivered by the special educator. All participants underwent pre- and post-intervention evaluation for outcome measures. (3) Results: 40 children (NAO = 19, control = 21, similar baseline characteristics) were included. Pre- and post-intervention evaluation showed comparable improvements in both groups in cognition skills (decoding, phonological awareness and reading comprehension), while between-group changes favored the NAO group only for some phonological awareness exercises. In total, no significant changes were found in any of the groups regarding the emotional/behavioral secondary outcomes. (4) Conclusion: NAO was efficient as a tutor for a human-supported intervention when compared to the gold-standard intervention for elementary school students with SpLDs.

## 1. Introduction

Learning disorders are the most frequent neurodevelopmental disorder encountered in the setting of a typical education classroom, reflecting a broad spectrum of difficulties in reading, mathematics and writing expression [1]. Terms and definitions (dyslexia, dysgraphia, dysorthography, dyscalculia, combined learning disorders, etc.) have been used quite arbitrarily during the past 30 years with the biggest part of the scientific community currently adopting the approach of the fifth edition of the *Diagnostic and Statistical Manual for Mental Disorders* [2]. The authors of the manual proposed the umbrella term of “Specific Learning Disorder”, which refers to: any persistent difficulty in reading, writing, arithmetic or mathematical reasoning skills; starting at school-age years; resulting in lower-than-average academic scores; interfering significantly with the person’s daily life; and not being better justified by any other neurodevelopmental impairment [3].

The prevalence of Specific Learning Disorders (SpLDs) is quite variable across studies depending on the samples or on the definition used [4]. The estimated prevalence of dyslexia (the most common learning disorder) in early studies was around 3–7% in the general population [5], while more recent studies including different types of SpLDs estimate a prevalence of around 15% in the school-age population [6]. A study in a German student population of third- and fourth-grade elementary school students has shown that about half of children with an SpLD have problems in other learning domains as well [7]. This is extremely important for developing diagnostic and intervention strategies for this population, which is even nowadays frequently underdiagnosed or late-diagnosed [8].

In fact, early diagnosis and intervention are crucial for achieving the optimal academic outcome of children with SpLDs [9]. Although many propositions and instructions for optimal management are available, there is to date no gold-standard intervention strategy used in everyday educational or research settings [10]. Intervention models in dyslexia, for example, focus on a group of impaired cognition domains, such as phonological awareness, working memory and reading efficiency; all those three abilities are complementary to one another, and an effective intervention involves parallel training in all three different abilities [11]. Other interventions focus on ameliorating decoding (identification of spelling patterns, word families, root words and phoneme segmentation). Decoding is significantly correlated to reading comprehension, especially in young children. Decoding and reading comprehension are among the most important components of reading achievement [12].

Furthermore, the interventions for SpLDs should also consider the potential effect on the emotional/behavioral status of the children as problems in several domains have been identified in this group (i.e., internalizing and externalizing problems such as anxiety, depression, attention, etc.) [13]. The concomitance of learning disabilities and emotional/behavioral disorders along with the lack of a conceptual model and research focus on students that might present symptoms of both disorders has been well-established for more than 20 years now [14]. More recent studies have further confirmed this association. A study investigating the social, emotional and behavioral challenges in a Turkish student population with SpLDs identified abnormal total scores in the “Strengths and Difficulties Questionnaire” in around half of the study’s population, further correlating this percentage with other socioeconomical factors [15]. *Castro et al.* also showed a positive correlation with the presence of emotional–behavioral difficulties and the risk of worsening reading and math performance with a moderate worsening effect of female sex. The authors concluded that programs targeting learning difficulties should take into consideration the complexity of this association [16].

At the same time, artificial intelligence and robotics have progressively been integrated in human daily life with several applications in educational settings focusing on typical students or on those with special education needs [17]. Experiments using robots as a teacher, a companion or a learner have shown encouraging results in improving different learning abilities, such as vocabulary learning, language learning, reading and handwriting [18,19,20,21]. A recent review of the usage of SRs in learning has concluded robots show real potential in the field due to their ability to enhance learning with kinesthetic interaction, adaptive empathetic feedback and individualized adaptation of the content to the pupil and by improving motivation, engagement and self-esteem [21]. *Johal* also highlights the evolution in recent experiments towards the adoption of measures of efficacy that evaluate both cognitive outcomes (i.e., a score for specific skill performance) and affection outcomes (i.e., children/parents/educators or even robots’ reports on engagement, attention, reactivity, behavior, etc.) [21].

A meta-analysis evaluating the overall cognitive effect size of robot tutoring in educational settings (using Cohen’s d weighted by N) was 0.70 [95% confidence interval (CI), 0.66 to 0.75], while the aggregated mean affective outcome effect size was calculated as 0.59 (95% CI, 0.51 to 0.66) [22]. According to this paper, the cognitive effect size of the robot tutoring was comparable to the one of the gold-standard human tutoring (0.79), as reported in former studies. However, the authors concluded there was a lack of human–robot comparative studies that could further confirm this [22].

Similarly, in a review conducted by our team focusing on special education only, experiments with SRs including children with neurodevelopmental disorders (mainly represented by autism spectrum disorders or multiple difficulties) have mostly shown positive effects on cognitive and learning outcomes as well as behavioral and social skills. No safe results regarding the efficacy of the SR in special education could be derived as most of the available studies are not comparable; the sample sizes and methodology of most studies are poor; the intervention designs, outcome measures and comparators (when present) are not consistent across studies; and the interventions are not designed to target the specific educational needs of the populations under investigation [23]. Additionally, SpLDs are not adequately represented amongst the most frequently investigated neurodevelopmental disorders [9,23].

In this view, we aimed to investigate the efficacy of a social robot as the tutor with the assistance of a special educator for the delivery of an intervention carefully designed to address the specific needs of children with a SpLD. The potential effect of the intervention was evaluated using both cognition and affection outcomes and compared to the gold-standard delivery of the same program by a special educator/tutor without the participation of a social robot in the educational procedure.

## 2. Materials and Methods

A prospective randomized case–control study was conducted as part of the project SRTSE: Social Robots as Tools in Special Education (project’s website: http://humain-lab.cs.ihu.gr/index.php/portfolio-item/koiro3e/?lang=en, accessed on 5 June 2022). The study was conducted between March 2019 and October 2021.

### 2.1. Recruitment and Randomization

We addressed an open invitation for SpLD evaluation and possible participation in the study with a short description of the project to families and educational facilities in the Thessaloniki area (Greece) via social media platforms, the local press and the website of “Papageorgiou Hospital”. The evaluation process was addressed to children in the 3rd or 4th grade of elementary school that already had a diagnosis of a Special Learning Disorder (dyslexia, dysgraphia, dysorthography) or that were struggling at school without an official evaluation/diagnosis.

After an initial screening by telephone to all the interested candidates, we invited all children whose parents/tutors/educators had indicated interest and potential eligibility to provide a detailed history and to attend a first assessment by a specialized neuropsychologist (and a pediatric neurologist if indicated) at General Hospital Papageorgiou. A total of 134 students (89 boys and 45 girls) in the 3rd and 4th grade of elementary school completed the initial evaluation. During this evaluation, proof and official learning tests were collected in order to evaluate the accuracy of the diagnosis and the specific learning difficulty profile of all children with an already established diagnosis of a SpLD from an official and qualified organization. All children with a high suspicion but without an official diagnosis of a SpLD were further recommended to the outpatient clinic for learning difficulties at the Psychology Department of G.H. Papageorgiou in order to establish a possible diagnosis of an SpLD. The diagnostic procedure was based on an extended evaluation by a special education teacher and a neuropsychologist at the learning difficulties outpatient clinic at the Psychology Department of G.H. Papageorgiou and included:Obtaining a detailed history from the parents looking for important indications of an SpLD;Receiving and reviewing comments from teachers indicating the presence of an SpLD persisting for at least 6 months;Confirmation of deviation of 1.5 standard units for age in at least one test/subtest of the administered batteries during the learning assessment;Evaluation of the clinical picture combining all of the above along with other clinical observations.

All patients with a definite diagnosis of a SpLD subsequently underwent an intelligence test (WISC-III, WISC-V, RAVEN) unless they could provide the results of an official intelligence test in the two preceding years [3].

All children that completed the aforementioned evaluation and fulfilled the inclusion criteria were included in the study. Inclusion criteria were: (1) diagnosis or confirmation of diagnosis of an SpLD during the initial evaluation; (2) educational placement at the 3rd and 4th year of an elementary typical-education school without special education support during the program’s duration; (3) total intelligence quotient > 90; (4) absence of neurological (i.e., motor or sensory problems that may affect functionality, epileptic disorders), neuropsychiatric or neurodevelopmental comorbidities (children with attention-deficit and hyperactivity disorder were not excluded); (5) absence of special phobias related to the use of technology and contact with robots; and (6) inability to commit to participation in the intervention for 3 months (i.e., long distance to intervention’s location, parental lack of time). Informed consent was obtained from all subjects that participated in the study represented by their legal tutors.

### 2.2. Randomization and Intervention

After the initial assessment, the study finally included forty children. Participants were randomly allocated to two groups:Robot group. The social robot NAO was used as a tutor assisted by the special educator teacher for the delivery of a program of learning intervention with various educational scenarios.Control group. Participants received the same learning intervention program delivered by the special education teachers without a social robot.

The social robot NAO was selected for the robot-tutored sessions (Figure 1). The applicability and effectiveness of this robot have been well-established after more than ten years of human–NAO interaction and around 300 research works reporting its usage in several different daily life settings [24]. NAO is also frequently used in educational robotics research settings [21]. In our study, robot selection depended on two main factors: (a) its humanoid form and small size (57 cm in height with a head, arms, legs and torso), which makes it appealing to children, and (b) its extensive interaction capabilities. The latter include vision, face tracking for maintaining eye contact, text-to-speech function, speech recognition, sound detection and head, arm and leg motion, as well as LEDs and sound playback to provide visual and auditory stimuli and feedback. NAO’s role during the educational sessions was to initiate and end the sessions with welcome and goodbye messages, to deliver instructions for activities, to provide further explanation or help in the case of severe difficulties during the activity, to give feedback (positive and negative) according to the student’s performance after the completion of each activity and to engage interest and attention during and between activities thanks to its multisensory features.

During the therapeutic sessions, the robot also used its camera and microphones to record certain parameters relevant to the assessment of the child’s performance in the various activities defined in the protocol. The recorded data were stored in text files, allowing a potential retrieval, analysis and comparison with the clinical data. The implementation of the intervention program was assisted by a special educator, who intervened only when necessary for the smooth conduct of the intervention program as defined by the research protocol to enhance as much as possible the robot–child interaction. Details of a feasibility study that preceded the design of the interventional scenarios and the technical details of the measurements used to evaluate child–robot interaction during the educational scenarios have been published elsewhere [25].

The intervention followed a 1:1 intensive special education program delivered in twenty-four sessions with a frequency of two sessions per week. The duration of each session ranged from thirty to forty-five minutes. The educational scenarios had a special focus on reading decoding (through phonemic awareness and other exercises) and reading comprehension skills. Each session included 4–6 activities of graded difficulty that were designed based on teaching cognitive and metacognitive strategies. Some examples of those activities included: the reading of a predefined text evaluating the speed and number of errors, phonological awareness activities evaluating the number of right and wrong answers in the synthesis/segmentation/addition of predefined words, decoding of story texts, sentence synthesizing, word memory and other. The working scenarios were all similar in content, structure and succession for both the NAO and the control group with the only deference that the welcoming, the instructions, the support and the feedback for the activities was delivered by the special educator for the control group.

Interventions for both groups took place in a room situated in the facilities of the Children & Parent Center S.A, which was specifically designed as an appropriate room with a minimum distracting learning environment and at the same time suitable for the use of the robot. The intervention was conducted by an experienced special education teacher that was employed full-time by the aforementioned projects’ partners. The special educator observed and recorded the course of the intervention and the performance of the children for each performed activity in the special leaflets designed for this purpose for both groups. He also had the obligation to inform the parents and guardians about children’s performances based on those observations after the end of each session.

### 2.3. Outcome Measures

The effect of the intervention was measured and compared within and between groups based on an evaluation that was conducted at the beginning (T0) and at the end of the intervention by a neuropsychologist blinded to the allocation group of each student. The evaluation measures were defined with the cooperation of the multidisciplinary team of the outpatient clinic for learning difficulties and included a carefully chosen combination of tests/subtests of several available and validated learning assessment batteries and questionnaires for the Greek school-age population. The primary outcome measures included performance scores on different reading and writing activities of the Language Proficiency Criterion (L-a-T-o), the Detroit Test of Learning Aptitude (4th Edition, DLTA-4) and the Kaufman Test of Educational Achievement scale (3rd Edition, KTEA-3).

Additionally, parents were asked to complete the Child Behavior Checklist (CBCL) of the Achenbach System of Empirically Based Assessment (ASEBA) and the Strengths and Difficulties Questionnaire (SDQ).

The observations of the special educator with regard to participants’ performance per activity and per session were also evaluated as a secondary outcome reflecting the course of the intervention for both groups.

Finally, at the end of the intervention (T1), the parents and the children of both groups completed a satisfaction questionnaire.

### 2.4. Data Analysis

All statistical analyses were performed using SPSS 24 (SPSS, Chicago, Illinois). As the number of participants per group was small and not equal between groups, we performed non-parametric statistical tests. We proceeded with a by-group comparison of changes (baseline vs. end of treatment) for all scores using the paired-sample Wilcoxon test (a non-parametric equivalent of a paired-difference test, such as the paired Student’s *t*-test). The comparisons of changes between the two groups (control vs. NAO group) were calculated with the Mann–Whitney independent sample tests (a non-parametric alternative to the independent Student’s *t*-test). The standardized mean difference (Cohen’s d) was also calculated in order to measure the effect size when there were significant differences in both groups.

## 3. Results

### 3.1. Participants

Forty children (mean age 8.73 ± 0.64, 80% male, 52.5% in the third grade and 47.5% in the fourth grade of primary school, 22.5% with at least one unemployed parent) with an IQ score > 70 (mean IQ score 99.52 ± 8.28) were enrolled in this study. The NAO group included 19 children (mean age 8.58 ± 0.61, 89.5% male, 52.6% in the third year of primary school, mean IQ score 100.89 ± 8.43, 52.6% with pre-existing diagnosis of SpLD), and the control group included 21 children (mean age 8.86 ± 0.66, 71.4% male, 52.4% in the third year of primary school, mean IQ score 98.29 ± 8.16, 52.6% with pre-existing diagnosis of SpLD). Children in the two groups did not statistically differ in age (*U* = 154.500, *p* = 0.174), sex (*χ*2(1) = 2.030, *p* = 0.154) or IQ quotient (*U* = 156.000, *p* = 0.238).

### 3.2. Pre- and Post-Intervention Evaluation

All pre- and post-intervention performance scores along with comparisons by group are presented in detail in Table 1.

After a paired comparison of the students’ performances between pre- and post-intervention, the results show a significantly better performance, less errors and faster speed in different subtests (L-α-Τ-o: reading speed and reading comprehension A; DLTA-4: reproduction of sentences and inversion of letters; KTEA-3: total flow of written speech and correct word flow of written speech) for the control group. Similar significant improvements were observed in the performance of subtests for the NAO group too (L-α-Τ-o: reading speed, total word production and sequence of sentences; KTEA-3: correct word flow in written speech).

No significant differences were found in the completion of the SDQ for the control group. Results show significant differences only between the subscale problems with peers, indicating lower means at the post-intervention evaluation for the NAO group.

Finally, for the CBCL of the Achenbach System of Empirically Based Assessment (ASEBA), results show significant differences only between the subscale “Thought problems”, indicating higher means at the post-intervention evaluation of the control group, while no significant differences were found for the NAO group.

The effect size for the subtests with significant differences for both groups was calculated. This was the same for the L-a-T-o reading speed difference between the groups (dz* = 0.91), showing a strong effect for both of them. The effect size for the KTEA-3 correct word flow of written speech subtest was greater in the control group (dz* = 0.81 vs. 0.75), showing a greater benefit for the control group in this subtest.

Differences between the two groups both at baseline and at the end of treatment were also investigated. No significant differences were found between the two groups at the pre- and the post-evaluation (for L-α-Τ-o, DLTA-4, KTEA-3, SDQ and CBCL of the Achenbach System of Empirically Based Assessment (ASEBA), all *p* > 0.05).

### 3.3. Phonological Awareness and Reading Comprehension (Educator Scores)

Concerning the secondary results of the intervention based on the notes of the special educator, we grouped the results into two basic categories: phonological awareness and reading comprehension.

For phonological awareness, we compared activities involving synthesis, segmentation of speech and phoneme addition at the beginning and at the end of treatment for both groups. Results show statistically significant differences in phonological awareness concerning the control group only for the phoneme addition total correct and total wrong answers (*Z* = 2.821, *p* = 0.005 and *Z* = −2.058, *p* = 0.040), showing more correct and less wrong answers at the end of treatment. For the NAO group, statistically significant differences were shown in the segmentation total correct and total wrong answers (*Z* = 2.838, *p* = 0.005 and *Z* = −2.537 *p* = 0.011) and phoneme addition total correct answers (*Z* = 2.417, *p* = 0.016), also indicating more correct answers at the end of treatment (Figure 2). The effect size was greater for the control group between the beginning and the end of treatment (dz* = 0.71 vs. 0.63) concerning the phoneme addition correct answers.

For reading comprehension, we compared activities involving comprehension at the beginning, at the middle and at the end of treatment. For both groups, there was a statistically significant difference between the beginning and the middle of treatment, with the end of treatment (control group: *Z* = 3.442, *p* = 0.001 between the beginning and the end of treatment; *Z* = 3.656, *p* > 0.001 between the middle and the end of treatment; NAO group: *Z* = 2.809, *p* = 0.005 between the beginning and the end of treatment; *Z* = 3.553, *p* > 0.001 between the middle and the end of treatment) showing more correct answers as the treatment continued (Figure 3). The effect size was greater for the control group between the beginning and the end of treatment (dz* = 1.50 vs. 0.82) but greater for the NAO group between the middle and the end of treatment (dz* = 2.54 vs. 3.58).

Concerning the results of the notes of the special educator during the intervention (for phonological awareness: activities involving synthesis, segmentation of speech and phoneme addition and for reading comprehension: activities involving comprehension of texts), statistically significant differences were observed between the two groups at baseline and at the end of treatment only for phonological awareness. More specifically, the NAO group had more correct answers and fewer errors in some activities (Table 2).

### 3.4. Satisfaction

Finally, concerning satisfaction with participating in this study, we measured parents’ as well as children’s total satisfaction. Results show a high satisfaction with a mean “Total satisfaction” score for children of 4.08 (±0.97, MIN: 1, MAX: 5) and 7.97 (±1.58, MIN: 5, MAX: 10) for parents. More specifically, the mean total satisfaction score of parents and children in the control group was slightly higher than those in the NAO group (mean score 8 vs. 7.9 for parents and 4.23 vs. 3.84 for children), but this difference was not statistically significant (*p* > 0.05).

## 4. Discussion

The results presented above support the efficacy of the social robot NAO as a tutor in a human-assisted special education program for children with SpLDs. Social robots have been widely used in educational procedures with proof of superiority compared to tele-present and virtual agents [26]. Studies investigating their applicability and efficacy in special education and especially in SpLDs are thus of high importance as evidence on the field is lacking [27].

Our results support comparable effects between human and NAO tutoring in the SpLD education setting, with the benefit of evaluating both cognitive and affection outcomes. Both groups showed significant improvement in some of the reading, decoding and writing skills tested. The effect size of the improvement in reading speed was identical between the groups (dz* = 0.91) and comparable to that in the correct word flow of written speech (dz* = 0.81 (control group) vs. 0.75 (NAO group)). Additionally, there was no deterioration of any skill/subtest after the intervention for any of the groups, reflecting the success of the intervention for both groups. Finally, between-group comparisons for the changes in each skill did not significantly differ between the two groups, showing the NAO tutoring was as efficient as the human-tutored intervention.

The random allocation of the participants to the two groups at baseline along with the blinded pre/post-intervention evaluation of the above cognitive outcomes offers an added value to these results. Although the two groups were quite homogenous after randomization, the percentage of boys in the sum of participants was quite high (75.7%). According to large epidemiological studies across the world regarding gender differences in SpLDs, the rates of reading disability are higher in boys with the highest reported odds ratio at 3.19 (95% confidence interval (CI), 2.15–4.17) [7,28]. The increased male prevalence could be the explanation for the observed imbalance in our sample. Although some studies have shown differences in human–robot interactions between boys and girls, there is to date no study with sufficient evidence and/or a sufficient effect size supporting different efficacies of SRs based on participants’ gender [29].

The above results of our study were further confirmed by the performance scores of each activity as reported by the special educator during the course of the intervention for both groups. More specifically, phoneme addition total scores were improved at the end of the program with similar effect sizes for both groups, while the NAO group also showed better scores at the end of the intervention in the segmentation exercises. Moreover, comparisons between the groups have shown that the robot-tutored group, which already had significantly more correct answers in the phonological awareness segmentation exercises at baseline, sustained a significantly higher mean score of correct answers during the course of the intervention and also achieved better scores for the synthesis exercise at the end of the intervention. The effectiveness of robot-assisted training in improving phonological awareness in children with reading disabilities has already been reported in a previous Korean study with significant differences across the test periods for the number of correct responses for words with phonological rules of invented spelling for both the robot-assisted and control group [30]. These results are quite encouraging as phonological awareness is one of the most well-established predictors of dyslexia and reading capacities [31,32].

A few teams have reported positive results of an SR intervention for improving the skills of children with dysgraphia in isolated cases [33]. The IRECHECK project is an ongoing larger study on the same topic that will use a “learning-by-teaching” SR intervention aiming to improve writing skills [34]. Improvements in the writing skills were observed in our sample too, although the intervention and evaluation were not specifically focused on this skill. Our intervention and study design aimed in fact to evaluate the overall and segmented efficacy of the SR NAO in improving the learning abilities of our sample.

The ROBIN project for the use of a robotic platform in dyslexia-affected pupils also adopted a more multi-faceted intervention with several activities involving memory and decoding, visual–spatial understanding and reading comprehension [35]. Reading comprehension was in our study progressively ameliorated for both the NAO and the control group. Both groups had a relative decrease in their scores at the middle of the intervention, probably associated with the increase in the level of difficulty of the educational scenarios at that point. At the end of the intervention, reading scores were higher compared both to the middle and the beginning of the treatment with no significant differences in the positive effect between the two groups.

Finally, efficacy of the intervention at the assessment of qualitive/affection outcomes as measured by the widely used CBCL and SDQ questionnaires was much less prominent, with no deterioration observed in any of the assessed items and with no difference in the responses compared to the control group. Significant improvement was only seen in the item of problems with peers (SDQ) for the NAO group, showing fewer problems at the end of the intervention. Many studies on robot-assisted learning have shown promising results on different affection outcomes (engagement, attention, social behaviors, etc.), especially when the robot used had a humanoid appearance [36]. Although absolute scores for difficulties and problematic behaviors were lower for the NAO group, none of these differences were statistically significant, probably reflecting the need for bigger samples to identify subtle improvements associated with human–robot interventions in these important-for-daily-life-quality functions of children with SpLDs.

The satisfaction was very high for both groups, and the rates did not statistically differ between groups. When comparing the absolute satisfaction rates, especially according to the children’s questionnaires, the control group rated the intervention higher (4.23 vs. 3.84/5). This difference, although statistically non-significant, probably reflects one of the main limitations of the study, which was linked to some technical issues encountered by the NAO group during the interventions.

We tried to maximize the robot’s autonomy with adjustments following the feasibility study [25], but we did not manage to resolve all issues. The latter mainly included difficulties in children’s speech recognition. The special educator would intervene by giving NAO the appropriate instructions (i.e., repeat instruction, give positive feedback), remotely or by touching specific parts of the robot, only when needed (i.e., in case the child’s answer was not recognized). This could occasionally cause disturbance and disappointment to the child, but the special educator would still not take NAO’s place in tutoring so as to minimize the effect of the educator during the SR-tutored intervention. Speech recognition, especially in the setting of interaction with children, has already been identified as one of NAO’s weaknesses [24]. Integration of more advanced speech recognition softwares to complete/ameliorate NAO’s speech recognition could be considered for future works, especially for children and for populations without normal social and communication abilities. A further increase in the robot’s autonomy (or impression of autonomy) could also be encountered via distance supervision of the educational procedure and remote control of the robot (when needed) by the educator without their physical presence in the education room.

Another limitation was the small number of participants. We designed a specific-for-SpLDs long-term intervention, and the work-load of a bigger sample could not have been supported with the given resources. We cannot exclude a selection bias favoring males that could have been avoided with a bigger random sample. Patient enrollment and completion of the intervention was further complicated by the COVID-19 pandemic. Unexpected delays have not permitted further evaluation of the students’ performances at later time points (i.e., 3 months) after the end of the intervention. The long-term efficacy of every educational intervention is of high importance and rarely evaluated in SR experiments.

In conclusion, children with SpLDs are a quite distinct category of students in need of special education. They represent a normal-intelligence population, most frequently with no other comorbidity, that are part of the typical education population, but they have very specific difficulties and individualized re-educational needs. Our study supports that a tutored intervention delivered by the humanoid robot NAO, with a particular focus on decoding exercises, phonological awareness and reading comprehension, improved specific skills of the aforementioned areas of focus in elementary school children with SpLDs. The intervention tutored by the humanoid robot NAO was as effective as the gold-standard human-tutored intervention at improving reading abilities, while there was no significant impact of the interventions (with or without the robot) on the students’ social and behavioral status. Larger comparative studies and further development of the human–robot interaction capacities could establish SRs as important tools in special education for children with SpLDs.

## Figures and Tables

**Figure 1 children-09-01155-f001:**
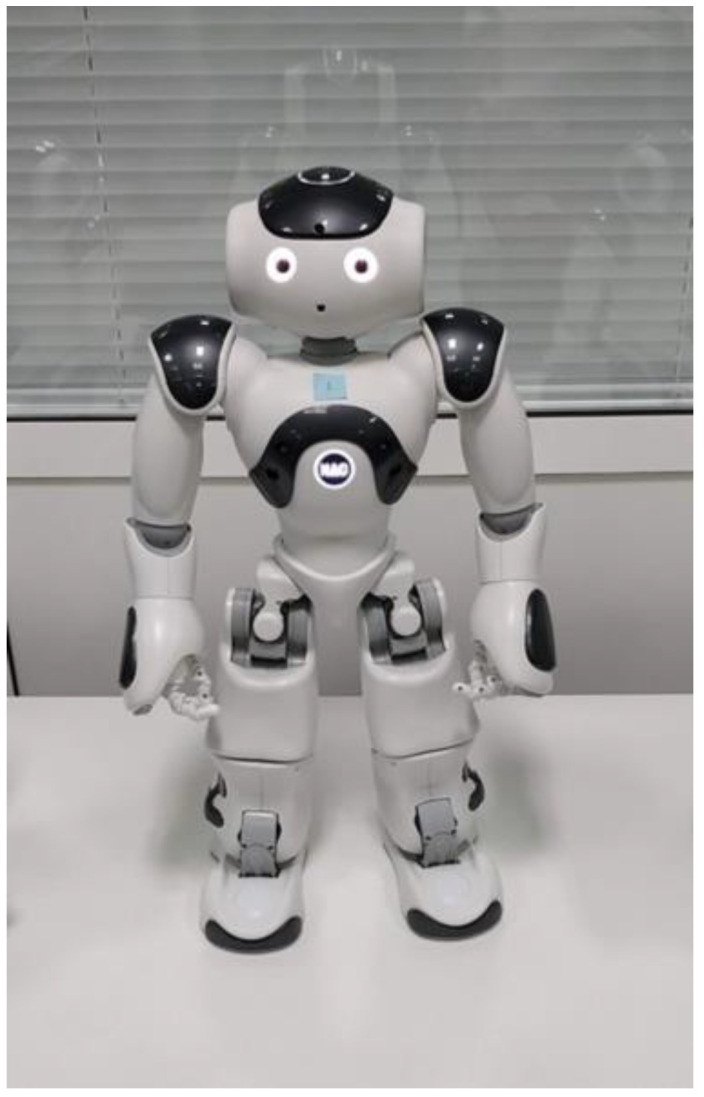
The humanoid social robot NAO.

**Figure 2 children-09-01155-f002:**
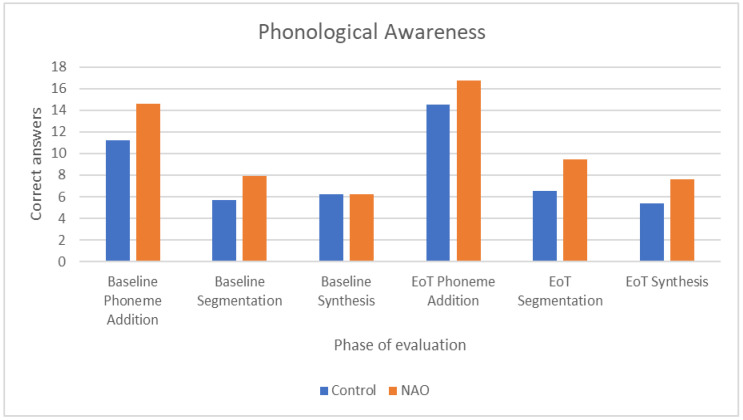
Correct answers in phonological awareness tasks between baseline and end of treatment (EoT) by group.

**Figure 3 children-09-01155-f003:**
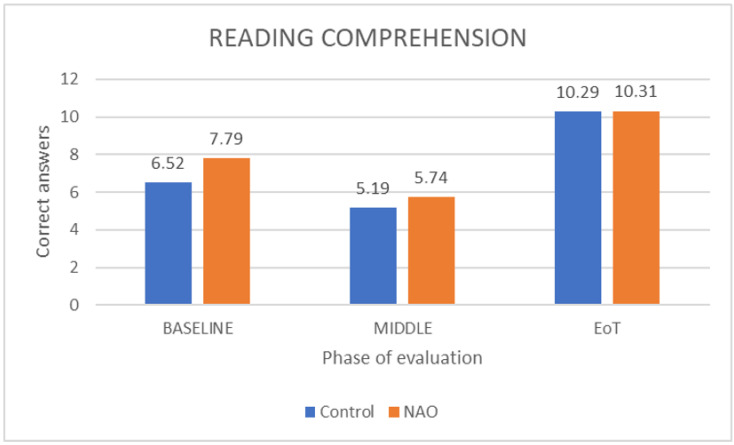
Correct answers in reading comprehension at the beginning (baseline), at the middle and at the end of treatment (EoT) for both groups.

**Table 1 children-09-01155-t001:** Pre- (1st) and post- (2nd) intervention evaluation by group.

Assessments	*Group*	*N*	*1st Evaluation Mean* *(±SD)*	*2nd Evaluation Mean* *(±SD)*	*Z*	*p*
L-α-Τ-o Reading Speed	CT	21	217.48 (±76.06)	174.90 (±61.29)	−3.024	0.002
	NAO	19	252.58 (±111.45)	186.47 (±88.67)	−3.662	>0.001
L-α-Τ-o Story Recognition and Understanding	CT	21	3.19 (±0.98)	3.62 (±1.24)	−1.277	0.202
	NAO	19	3.53 (±1.22)	3.79 (±1.27)	−1.072	0.284
L-α-Τ-o Total Word Production	CT	21	28.29 (±17.40)	31.76 (±19.18)	−0.887	0.375
	NAO	19	19.32 (±13.50)	25.37 (±13.43)	−2.113	0.035
L-α-Τ-o Correct word Production	CT	21	18.52 (±12.71)	19.95 (±14.19)	−0.654	0.513
	NAO	19	13.42 (±13.53)	16.74 (±12.07)	−1.824	0.068
L-α-Τ-o Reading Comprehension A	CT	21	1.33 (±1.39)	2.90 (±2.17)	−2.641	0.008
	NAO	19	2.42 (±2.46)	2.89 (±2.18)	−0.737	0.461
L-α-Τ-o Reading Comprehension B	CT	21	4.95 (±2.09)	5.62 (±1.63)	−1.175	0.240
	NAO	19	4.05 (±2.27)	5.00 (±2.26)	−1.460	0.144
L-α-Τ-o Sequence of Sentences	CT	21	1.67 (±1.98)	2.24 (±2.63)	−1.109	0.268
	NAO	19	1.53 (±1.74)	2.74 (±2.47)	−2.267	0.023
DLTA−4 Reproduction of Sentences	CT	21	13.86 (±5.83)	16.10 (±4.53)	−2.007	0.045
	NAO	19	14.42 (±6.20)	15.11 (±6.28)	−0.390	0.697
DLTA−4 Inversion of letters	CT	21	2.14 (±0.91)	3.43 (±1.91)	−2.882	0.004
	NAO	19	2.11 (±1.41)	2.84 (±2.93)	−0.862	0.389
KTEA−3 Total flow of written speech	CT	21	37.62 (±13.37)	46.52 (±15.62)	−3.121	0.002
	NAO	19	39.42 (±18.12)	41.74 (±19.01)	−0.900	0.368
KTEA−3 Correct word flow of written speech	CT	21	25.52 (±11.06)	30.57 (±13.16)	−2.244	0.025
	NAO	19	23.32 (±13.73)	27.68 (±14.95)	−2.071	0.038
KTEA−3 Complete sentence flow in written speech	CT	21	8.62 (±6.63)	9.14 (±6.61)	−0.526	0.599
	NAO	19	9.42 (±6.35)	11.16 (±9.70)	−0.712	0.476
KTEA−3 Correct answer flow in silent reading	CT	21	14.29 (±4.21)	14.95 (±5.64)	−0.760	0.448
	NAO	19	12.74 (±6.19)	12.68 (±6.20)	−0.144	0.909
KTEA−3 Wrong answer flow in silent reading	CT	21	1.48 (±1.08)	1.43 (±2.60)	−0.953	0.340
	NAO	19	1.26 (±1.49)	1.05 (±1.27)	−0.447	0.655
SDQ Emotional problems	CT	21	2.29 (±1.68)	2.14 (±1.56)	−1.134	0.257
	NAO	19	2.68 (±2.19)	2.11 (±1.82)	−1.339	0.181
SDQ Communication problems	CT	21	3.05 (±2.27)	3.05 (±2.27)	−0.439	0.660
	NAO	19	2.58 (±2.17)	2.11 (±1.99)	−1.218	0.223
SDQ Hyperactivity	CT	21	4.90 (±2.90)	5.57 (±2.89)	−1.190	0.234
	NAO	19	5.11 (±2.67)	4.68 (±2.50)	−1.204	0.229
SDQ Problems with peers	CT	21	2.24 (±2.17)	2.19 (±2.04)	−0.632	0.527
	NAO	19	2.26 (±1.85)	1.68 (±1.77)	−1.995	0.046
SDQ Prosocial problems	CT	21	7.95 (±1.47)	7.71 (±1.38)	−1.000	0.317
	NAO	19	8.26 (±1.94)	7.89 (±1.73)	−0.710	0.478
SDQ Total problems	CT	21	12.76 (6.31)	13.24 (±6.14)	−0.259	0.795
	NAO	19	12.63 (±6.30)	10.84 (±5.03)	−1.374	0.169
CBCL Anxiety/Depression	CT	21	4.00 (±4.21)	3.67 (±3.55)	−0.320	0.749
	NAO	19	3.84 (±2.91)	3.79 (±2.88)	−0.686	0.493
CBCL Withdrawal/Depression	CT	21	2.48 (±2.36)	2.05 (±2.52)	−1.211	0.226
	NAO	19	2.95 (±2.42)	2.68 (±2.58)	−1.121	0.262
CBCL Somatic complaints	CT	21	0.86 (±1.39)	.95 (±1.32)	−0.073	0.942
	NAO	19	0.58 (±0.84)	0.58 (±0.77)	0.000	1.000
CBCL Social problems	CT	21	4.19 (±3.22)	4.38 (±3.34)	−0.323	0.747
	NAO	19	4.26 (±4.11)	4.16 (±3.80)	−0.159	0.874
CBCL Thought problems	CT	21	1.95 (±2.46)	2.48 (±2.50)	−2.047	0.041
	NAO	19	1.89 (±1.70)	1.84 (±1.54)	−0.322	0.748
CBCL Attention problems	CT	21	7.10 (±3.49)	7.33 (±3.81)	−0.343	0.732
	NAO	19	6.95 (±2.80)	6.58 (±3.12)	−1.485	0.138
CBCL Violation of rules	CT	21	2.48 (±2.94)	2.67 (±2.94)	−0.156	0.876
	NAO	19	2.00 (±2.77)	2.16 (±2.57)	−0.791	0.429
CBCL Aggressive behaviour	CT	21	6.10 (±5.92)	5.71 (±6.08)	−1.287	0.198
	NAO	19	4.26 (±4.80)	3.95 (±3.39)	−0.463	0.643
CBCL Other problems	CT	21	2.76 (±2.81)	3.38 (±2.64)	−1.109	0.267
	NAO	19	3.11 (±3.35)	3.26 (±3.05)	−0.516	0.606
CBCL Internalization	CT	21	7.33 (±6.45)	6.43 (±6.19)	−1.435	0.151
	NAO	19	7.26 (±4.71)	6.95 (±5.19)	−1.089	0.276
CBCL Externalization	CT	21	8.57 (±8.54)	8.76 (±8.80)	−0.649	0.516
	NAO	19	6.26 (±6.05)	6.05 (±4.34)	−0.090	0.928
CBCL Total	CT	21	31.76 (±23.15)	33.05 (±23.70)	−0.071	0.943
	NAO	19	29.79 (±17.40)	28.53 (±16.35)	−0.774	0.439

Notes: (N = number of subjects, CT = control group, NAO = NAO group, SD = standard deviation, Z = Z of Wilcoxon rank test for comparison of two means, *p* = *p*-value, L-a-T-o = Language Proficiency Criterion; DLTA-4 = the Detroit Test of Learning Aptitude, 4th edition; KTEA-3 = the Kaufman Test of Educational Achievement scale, 3rd edition; SDQ = Strengths and Difficulties Questionnaire; CBCL = Child Behavior Checklist of the Achenbach System of Empirically Based Assessment).

**Table 2 children-09-01155-t002:** Significant changes by group between the beginning (baseline) and the end of treatment (EoT) in the phonological awareness evaluation scores.

	Group	*N*	*M*	*SD*	Mean Rank	*U*	*p*
Total correct Baseline (segmentation)	NAO	19	7.95	1.985	24.89	116.000	0.023
	CONTROL	21	5.71	3.227	16.52		
Total correct EoT (synthesis)	NAO	19	7.58	2.009	26.18	91.500	0.003
	CONTROL	21	5.38	2.037	15.36		
Total wrong EoT (synthesis)	NAO	19	0.89	1.150	14.00	76.000	0.001
	CONTROL	21	3.00	2.258	26.38		
Total correct EoT (segmentation)	NAO	19	9.47	0.772	28.24	52.500	<0.001
	CONTROL	21	6.52	2.750	13.50		
Total wrong EoT (segmentation)	NAO	19	0.11	0.315	14.13	78.500	<0.001
	CONTROL	21	2.43	2.976	26.26		

Notes: (N = number of subjects, CT = control group, NAO = NAO group, SD = standard deviation, U = Mann–Whitney U-test for comparison of mean ranks, *p* = *p*-value of significance).

## Data Availability

The authors confirm that the majority of the data supporting the findings of this study are available within the article. Raw data are available from the authors upon reasonable requests.

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
