# Peer review of "Efficacy of a Robot-Assisted Intervention in Improving Learning Performance of Elementary School Children with Specific Learning Disorders"

_children, 2022, doi:10.3390/children9081155_

Round 1

Reviewer 1 Report

The study deals with an interesting topic: Robot, LD, and Learning.

In the introduction, a more comprehensive and detailed literature review would be appreciated by readers regarding the use of a social robot and its effectiveness in academic, social, and emotional outcomes of students with disabilities, especially students with LD. A robot was mentioned very briefly in the introduction (the last paragraph in the introduction), even though it is a major intervention tool.

The significance of contents needs a stronger foundation and justification for the study. The authors used a lack of studies on LD and robots as a justification. The authors can find justification of the use of social robots for students with LD from the findings of previous studies conducted with students with neurodevelopmental disorders.

The study indicated that their focus was on two areas (phonological awareness and reading comprehension). Using CBCL, social, behavioral, and emotional functions of students with LD were measured. The use of CBCL needs to be justified.

Moreoever, CBCL was only used for control groups. What was the reason?

The authors want to explain the reason for evaluating students' social, behavioral, and emotional performance, when the purpose of using the robot was to evaluate the performance on the phonological awareness and reading comprehension.

A detailed explanation about the social robot will be greatly appreciated: look, shapes, sizes, movement/actions of the robot, sounds, descriptions of interactions between a robot and a student with LD, etc. I somehow failed to find Picture 1, as it is neither in supplementary files nor in the manuscript. 

Why was the social robot NAO selected for the study? How was the effectiveness of this robot was reported in previous studies?

In the study, one special education teacher provided interventions to 40 students individually (e.g., 1:1) for 30 to 45 minutes two times a week at a hospital setting. It seems to be a very challenging intervention workload for one special education teacher. Was he a licensed special education teacher? Was he employed at the hospital full time? The authors want to provide a more detailed information about interventions and a description of a sample intervention.

In addition, explanations about a teacher-led intervention (control group) would be greatly appreciated.

In each group, how many students were previously diagnosed  with LD and potential LD?

There seems to be a discrepancy between the findings reported in different places: Abstract-"Within group changes showed comparable improvements in both groups in cognition skills (decoding, phonological awareness and reading comprehension); while between group changes favored the NAO group only for some phonological awareness exercises." VS. "Our study supports that a humanoid-robot NAO tutored intervention was effective on the improvement of reading abilities of elementary school children with a particular focus on decoding exercises, phonological awareness and reading comprehension." Please clarify.

Author Response

Please see the attachement. Thank you.

Reviewer 2 Report

See attached

ISSN 2227-9067

This manuscript describes findings of an intervention delivered by social robots (versus special education teachers) on a series of cognition skills in elementary school students with SpLDs. The results based between-group and between-group analyses revealed an overall effectiveness of the humanoid robot to a degree comparable to human tutors.

The paper is well written and addresses an important topic that is likely to interest many Children readers. My main concern regards the data analysis approach. It is not clear why the analyses/results were split into between-group and within-group changes. For mixed design experiments, between-group and within-group analyses should be integrated to generate the finding results by analyzing pre-test and post-test of both control and experiment group (i.e., four data points per participant) simultaneously. Separate analyses and emphases under different titles (in the results section) is not recommended. In addition, statistically, the results based on between-group and within-group data analyses should be consistent with each other. Thus, it is confusing to draw the seemingly different results that within group changes showed comparable improvements while between group changes favored the NAO group. I would suggest the authors to reconsider the analytical plan and adopt a more coherent analysis and report approach.

Other than that, I have only a few questions/suggestions for the author. In no particular order:

·       Please explain and cite how the non parametric statistical tests were performed, in terms of their underlying mechanics and calculations. Readers would benefits from more details.

·       Even though there is no statistically difference in sex, the number of male children in the samples account for 80%, which needs further explanation or review of relevant literature on the impact of having such an imbalanced sex in the results.

·       Please provide detailed statistics for statistical tests conducted for age, sex, and IQ between NAO group and the Control group.

·       Please double-check for missing statistics results for CBCL of the ASEBA for NAO group

·       Table S1 has to be restructured to present control group (pre vs post) and NAO group (pre vs post) for a specific assessment at the same time.

·       Table S1 and S2 need detailed denotation explaining the meanings of numbers and columns.

·       Visualizations (Figure S1 and Figure S2) need to be redesigned to better capture the comparison between groups through different colors or bar groups. It is hard to compare control and NAO group difference especially for Figure S2. There is also missing detailed denotations explaining meaning of numbers in the chart.

·       There is a decrease in the middle of treatment Reading comprehension, lack of explanations on why there are less correct answers compared with the beginning and end of treatment.

·       For between group changes, the authors need to show testing statistics in details in support of argument.

·       Participants were randomly allocated to two groups, but Table S2 shows there is a significant difference in Baseline (segmentation) at the beginning of treatment, thus it’s not clear the significant improvement for the Phonological awareness was simply due to the fact that participates in NAO group already had better performance.

·       Basic formatting issues. “Within-group changes” section is not numbered, but “Between-groups changes” section is labeled with “a”. Later on, the “satisfaction” section is also labeled with “a”.

·       In the discussion, the mention of comparable cognitive effect size to meta-analysis effect size doesn’t add any additional value, should be moved to the beginning of your paper.

·       When talking about satisfaction rates between the Control group and NAO group, the score for the Control group was greater than those of the NAO group. But later in the discussion, the opposite argument was indicating the satisfaction rates were higher for the NAO group.

·       The limitations of the study are under-discussed. Listing these limitations are not enough and potential mitigations solutions should be proposed so future studies know how to better avoid and collect better data.

·       The technical issues of the robot can make the results bias and can’t rule out the help of special educator entirely. This has to be addressed in details in terms of the effect, biases, and potential mitigation solutions going forward.

Round 2

Reviewer 1 Report

Thanks for the opportunity to review the revised manuscript.

The improved justification on the investigation of the emotional, social, and behavioral impact on students with LD when using a robot was provided in the revised version. Authors made the manuscript stronger by reorganizing their contents and showed their intention for their study at the beginning of the manuscript.

However, the justification does not seem to be sufficient enough to address social/behavioral challenges experienced by students with LD. The added one sentence (page 3, furthermore.....) cited only one previous study indicating the internalizing and externalizing behavioral issues of students with LD. Internalizing and externalizing behavioral issues are typically used to explain characteristics of students with emotional/behavioral disorders. 

In the revised manuscript, even though authors clarified that the purpose of the study was to address both focuses (learning and emotional/behavioral/social performance), the title, the abstract, and the conclusion have no mention about improvement/impact on social/emotional/behavioral performance. Authors need to conduct a more careful revision to explicitly present their multiple intentions/focuses (improving both learning performance and social/emotional/behavioral performance) in all major sections (title, abstract, and conclusion). 

A one-on-one intervention with the support of a special education teacher was explained in the original manuscript. In the revised manuscript, the author stated about interventions in a group of 10, following the one-on-one intensive special education program. I am very confused with the intervention method/sequence. Was it a group intervention? not a 1:1 intervention? How many robots were used? Was one robot used for a group of 10 students at the same time in the same room? Was a robot intervnetion happend after a one-on-one instruction/program with the special education teacher was provided? Please clarify. 

According to the results, the statisfically significant findings are found only in some scales. But the conclusion is plainly saying that the study supports the effectiveness of the humanoid robot NAO.  The conclusion statement seems to be too strong for the limited results this study obtained. 

Somehow, the computer I am using cannot open the .png files (supplementary files). 

Minor mechanical issues are found:

(1) Proper indentations at the beginning of all paragraphs are needed

(2) font styles/sizes are not consistent (page 3. The potential effect of the intervention...) 

Author Response

Please see attachement.

This manuscript is a resubmission of an earlier submission. The following is a list of the peer review reports and author responses from that submission.